# Integrated Multi-Omics Profiling Reveals That Highly Pyroptotic MDMs Contribute to Psoriasis Progression Through CXCL16

**DOI:** 10.3390/biomedicines13071763

**Published:** 2025-07-18

**Authors:** Liping Jin, Xiaowen Xie, Mi Zhang, Wu Zhu, Guanxiong Zhang, Wangqing Chen

**Affiliations:** 1Department of Dermatology, Xiangya Hospital, Central South University, Changsha 410008, China; jlp318@csu.edu.cn (L.J.); 2204140508@csu.edu.cn (X.X.); zhangmi_xy@csu.edu.cn (M.Z.); zhuwuxy@csu.edu.cn (W.Z.); 2National Clinical Research Center for Geriatric Disorders, Xiangya Hospital, Central South University, Changsha 410008, China; 3National Engineering Research Center of Personalized Diagnostic and Therapeutic Technology, Changsha 410083, China; 4Furong Laboratory, Changsha 410078, China; 5Hunan Key Laboratory of Skin Cancer and Psoriasis, Hunan Engineering Research Center of Skin Health and Disease, Xiangya Hospital, Central South University, Changsha 410008, China

**Keywords:** pyroptosis, psoriasis, CXCL16, monocyte-derived macrophages, multiomics

## Abstract

**Background:** Psoriasis, an inflammatory skin disorder, involves pyroptosis—a pro-inflammatory cell death process. However, cell-specific pyroptosis dynamics and immune microenvironment interactions remain unclear. **Objective:** To investigate cell-type-specific pyroptosis patterns in psoriasis and their immunoregulatory mechanisms. **Methods:** We integrated 21 transcriptomic datasets (from 2007 to 2020) obtained from the GEO database and two single-cell RNA sequencing datasets to quantify pyroptotic activity using Gene Set Variation Analysis and AUCell algorithms. Immune cell infiltration profiles were evaluated via CIBERSORT, while cell-cell communication networks were analyzed by CellChat. In vitro and in vivo experiments were performed to validate key findings. **Results:** Our analysis revealed that psoriasis patients exhibited significantly elevated levels of pyroptosis compared to healthy controls, with pyroptotic activity reflecting treatment responses. Notably, monocyte-derived macrophages (MDMs) in psoriatic lesions displayed markedly heightened pyroptotic activity. In vitro experiments confirmed that MDMs derived from psoriasis patients overexpressed pyroptosis-related molecules (Caspase 1 and Caspase 4) as well as pro-inflammatory cytokines (*TNFα*, *IL6*, *IL1β*) when compared to healthy controls. Furthermore, these cells showed increased expression of CXCL16, which might potentially activate Th17 cells through CXCR6 signaling, thereby driving skin inflammation. Inhibition of monocyte migration in an imiquimod-induced psoriasiform dermatitis model significantly alleviated skin inflammation and reduced the proportion of M1 macrophages and Th17 cells in lesional skin. **Conclusions:** This study revealed that MDMs in psoriatic lesions exhibited a hyperactive pyroptotic state, which contributed to disease progression through CXCL16-mediated remodeling of the immune microenvironment. These findings highlight pyroptosis as a potential therapeutic target for psoriasis.

## 1. Introduction

Psoriasis is a chronic, systemic inflammatory skin disease characterized by scaly, erythematous plaques or patches that may be localized or widely distributed across the body. Its pathogenesis is multifactorial, involving genetic, environmental, and immunological factors [1]. Accumulating evidence highlights the critical role of cell death in psoriasis progression [2,3,4,5]. Among the 11 recognized cell death modalities, pyroptosis—a highly inflammatory form of programmed cell death—plays pivotal roles in antimicrobial defense, tumorigenesis, and various autoimmune and inflammatory disorders [6]. Pyroptosis is initiated by inflammasome activation (e.g., AIM2, NLRP3) and proceeds via either the canonical Caspase 1 pathway or the non-canonical Caspase 4/5/11 pathway. This process is mediated by Gasdermin family proteins (GSDMA/B/C/D/E and DFNB59) [7], leading to plasma membrane rupture and the release of pro-inflammatory cytokines such as IL-1β and IL-18, thereby amplifying inflammatory responses [6].

Previous studies have revealed elevated expression of pyroptosis-related markers (e.g., GSDME, GSDMD, Caspase 3, IL1β, IL6, and TNFα) in psoriatic lesions compared to healthy skin or non-lesional areas [8,9,10]. Pharmacological inhibition of Caspase 3 or NLRP3, as well as genetic deletion of *Gsdme* or *Gsdmd* in murine models, confers protection against imiquimod (IMQ)-induced psoriasiform dermatitis [8,11], underscoring the pathogenic role of pyroptosis. Nevertheless, the role of pyroptosis in psoriasis remains incompletely understood. One study has reported that GSDMD is overexpressed in neutrophils within psoriatic lesions [9]. A single-cell RNA sequencing (scRNA-seq) analysis identified upregulated GSDME expression in keratinocytes from psoriasis patients [12]. In addition, TNF-α/caspase-3/GSDME- or GSDMD-mediated keratinocyte pyroptosis has been shown to exacerbate skin inflammation and disease progression [8,13]. Despite these findings, a comprehensive characterization of pyroptotic activity in psoriasis is still lacking.

In this study, we aim to systematically illustrate the role of pyroptosis in psoriasis by integrating bulk transcriptomic and single-cell multi-omics data. Specifically, we will define cell-type-specific pyroptotic signatures in psoriatic lesions and elucidate the mechanistic contributions of pyroptosis to disease pathogenesis. Our findings will not only advance our understanding of pyroptosis in psoriasis but also provide novel biomarkers and therapeutic targets for precision medicine.

## 2. Materials and Methods

### 2.1. Transcriptomic Data Preparation

Transcriptomic microarray datasets containing skin samples from patients with psoriasis vulgaris between 2005 and 2021 were downloaded from the Gene Expression Omnibus (GEO, https://www.ncbi.nlm.nih.gov/geo/, accessed on 22 October 2022), a public database hosted by NCBI. A total of 29 datasets were initially retrieved. After excluding datasets with fewer than 10 samples or incomplete data and annotations, 21 datasets were ultimately included (Appendix A). Transcriptomic data were log_2_(FPKM + 1)-transformed and harmonized using ComBat to correct for batch effects across platforms and studies.

### 2.2. Construction of a Psoriasis Pyroptosis Scoring Tool

Through the PubMed database of the National Library of Medicine and the Molecular Signatures Database (MSigDB, https://www.gsea-msigdb.org/gsea/msigdb, accessed on 29 August 2023), we systematically reviewed studies and reviews related to pyroptosis-related genes (PRGs) and identified 57 candidate PRGs (Appendix A). Baseline skin samples and normal control samples from all datasets were selected for differential expression analysis across 21 datasets using the “limma 3.56.2” R package, with screening criteria set at false discovery rate (FDR) < 0.05 and |log2FC| > 0.585. Dot plots and frequency bar charts generated using the “ggplot2 3.4.4” R package were generated to visualize expression variations and the distribution of significantly differentially expressed PRGs across datasets. PRGs exhibiting significant differential expression in ≥5 datasets were designated as the target gene set for subsequent analyses.

The non-negative matrix factorization (NMF) algorithm [14] was employed to identify distinct expression patterns of PRGs. Consensus clustering of expression pattern matrices was performed using the “ConsensusClusterPlus 1.64.0” R package, where the k-means clustering algorithm and Euclidean distance metric were applied. The optimal cluster number was determined through cumulative distribution function (CDF) analysis. This process yielded two pyroptosis-related gene clusters with strong intra-group correlations and weak inter-group associations. Gene Set Variation Analysis (GSVA) was conducted using the “GSVA 1.48.1” R package to quantify expression score matrices of both clusters across the 21 datasets. To validate which cluster better reflects pyroptosis status, GSVA scores were assessed in two independent pyroptosis-related datasets (GSE57253 and GSE153494) with inter-group differences analyzed by Kruskal–Wallis test. Through integration with original literature findings, the cluster effectively representing pyroptosis status was identified and designated as the “Psoriasis Pyroptosis Score”.

### 2.3. Analysis of Pyroptosis Scores and Psoriasis Characteristics

Logistic regression analysis was conducted to assess the impact of psoriasis-associated pyroptosis scores on the risk of psoriasis across the 21 datasets. Odds ratios (ORs) were computed to quantify the associated risks. Subsequently, receiver operating characteristic (ROC) curve analysis was performed using the “pROC 1.18.5” R package to determine the area under the curve (AUC) and evaluate the discriminatory power of pyroptosis scores in distinguishing psoriasis cases from healthy controls. Furthermore, a dataset comprising treatment response information (GSE11903 and GSE117239) was utilized to calculate the differences in focal pyroptosis scores before and after treatment with biologic agents (ustekinumab and etanercept). Additionally, ROC analysis was conducted with PASI 75 as the threshold to assess the efficacy of changes in focal pyroptosis scores in predicting treatment response.

### 2.4. Immune Cell Infiltration Assessment

Immune cell infiltration analysis was conducted on 21 GEO datasets utilizing the “CIBERSORT 0.1.0” R package. CIBERSORT estimates the enrichment levels of 22 types of human immune cells through linear support vector regression and deconvolution algorithms [15]. Pearson correlation analysis was employed to investigate the correlation between psoriasis scorch death scores and immune cell infiltration levels, calculate the correlation coefficient (r) and significance (*p* value), and identify the types of immune cells most associated with scorch death using the “ggplot2” R package for visualization [16].

### 2.5. Single-Cell Analysis of Psoriasis Datasets

Single-cell omics data (GSE121212) were obtained from the Zenodo research data platform (https://zenodo.org/records/4569496, accessed on 17 August 2023), which provides the pre-validated psoriasis atlas published by Reynolds et al. [17]. In this original study, cell clustering was performed using the Leiden algorithm with marker validation via CyTOF and immunohistochemistry, defining 41 transcriptional subtypes. Data quality control, standardization, and normalization were performed using the “Seurat 4.4.0” R package. Unified Manifold Approximation and Projection (UMAP) was applied for dimensionality reduction and visualization to identify cell subpopulations based on these annotated clusters. Given the limitations of GSVA analysis in single-cell applications, we employed the “AUCell 1.22.0” R package to evaluate cellular pyroptosis levels. Through comparative analysis of mean GSVA scores and mean AUCell values across cell subtypes, we demonstrated that AUCell effectively captures pyroptosis activity levels. The area under the curve (AUC) of pyroptosis-related gene sets was calculated using AUCell analysis, enabling bimodal threshold-based classification of cells into “high-pyroptosis” and “low-pyroptosis” states. These pyroptosis status differences across cell subtypes were visualized using the “ggplot2” R package. Subsequently, the “CellChat 1.6.1” R package was employed to construct cell communication networks and infer the communication patterns of intercellular signaling pathways.

Finally, parallel validation was conducted using the human psoriasis dataset GSE228421. We performed batch-corrected linear dimensional reduction in the coordinate space using Seurat, followed by unsupervised cell clustering implemented via the FindClusters function at a resolution parameter of 0.3. We then conducted subclustering analysis specifically for macrophages, with the optimal resolution parameter (0.5) determined through systematic evaluation using the “Clustree 0.5.0” package to ensure biologically meaningful cluster separation (Appendix A). Subsequent analyses focused on quantifying pyroptosis activation status in macrophages and evaluating the transcriptional activity of CXCL-linked signaling pathways.

### 2.6. Human Monocyte Culture and Differentiation

Human blood samples were obtained from healthy donors and psoriasis patients (Department of Dermatology of Xiangya Hospital, Central South University) with informed consent (Appendix A). This study was performed in accordance with the principles of the Declaration of Helsinki and approved by the Research Ethics Boards of Xiangya Hospital of Central South University (202308636). Human peripheral blood mononuclear cells (PBMCs) were first isolated using Ficoll-Paque (DAKEWE, Shenzhen, China; 7111011) density gradient centrifugation. Then, monocytes were cultured in RPMI 1640 (Gibco, Waltham, MA, USA; C11875500BT) supplemented with 10% FBS (ExCell Bio, Shanghai, China; 12J262) and 1% pen–strep (Biological Industries, Kibbutz Beit Haemek, Israel; 03-031-1B) in 5% CO_2_ at 37 °C. After 24 h, the non-adherent cells were removed, and the remaining adherent cells were further cultured in a medium supplemented with 50 ng/mL macrophage colony stimulating factor (M-CSF) (Peprotech, Cranbury, NJ, USA; 300-25). The culture medium was refreshed every two days. By day six of culture, monocyte-derived macrophages (MDMs) were successfully obtained [18,19].

To investigate the causal relationship between pyroptosis activation and CXCL16 expression, MDMs from psoriasis patients (*n* = 3) were treated with the pyroptosis-specific inhibitor VX-765 (Selleck, Houston, TX, USA; S2228) at a concentration of 40 μM based on prior dose-response optimization. Cells were exposed to VX-765 or DMSO vehicle control for 48 h in RPMI 1640 medium supplemented with 10% FBS and 50 ng/mL M-CSF, maintaining conditions identical to primary differentiation protocols.

### 2.7. RNA Isolation and Quantitative RT-PCR (RT-qPCR)

Total RNA from skin tissue or cells was isolated using MagZol reagent (Magen, Guangzhou, China; R4801-01) according to the manufacturer’s instructions. Reverse transcription was performed using the Hifair^®^ II 1st Strand cDNA Synthesis Kit (Yeasen, Shanghai, China; 11141ES60) according to the manufacturer’s instructions. Real-time PCR was performed using SYBR Green Master Mix (Bimake, Houston, TX, USA; B21703). The mRNA expression levels of the examined genes were normalized and then determined using the 2^−ΔΔCt^ method. Primer sequences are shown in Appendix A.

### 2.8. Western Blotting

Cells were lysed using RIPA buffer (Beyotime Biotechnology, Shanghai, China; P0013C) with a protease inhibitor cocktail (Bimake, B14002) and phosphatase inhibitor cocktail (Bimake, Houston, TX, USA; B15002). The antibodies used in this study included Caspase1 (Proteintech, Chicago, IL, USA; 22915-1-AP, 1:1000), Caspase4 (Proteintech, 11856-1-AP, 1:1000), GAPDH (Santa Cruz, Dallas, TX, USA; 60004-I-Ig, 1:3000), HRP goat anti-rabbit IgG (H+L) (ABclonal, Woburn, MA, USA; AS014, 1:5000), and HRP goat anti-mouse IgG (H+L) (ABclonal, AS003, 1:5000).

### 2.9. IMQ Model of Psoriasis

C57BL/6 mice were acquired from Hunan Slack King Experimental Animal Co. (Changsha, China). Age-matched and sex-matched mice were used for all experiments. All animal experiments were performed according to the principles specified in the “Guide for the Care and Use of Laboratory Animals in China” with approval from the Animal Ethics Committee of Central South University (CSU-2022-0692). Six- to eight-week-old C57BL/6 mice received a daily topical dose of 62.5 mg IMQ cream (Med·Shine, Chengdu, China; 41210301) on the shaved dorsal back for 4 days. An objective scoring system was used to assess skin inflammation based on the clinical Psoriasis Area and Severity Index (PASI) according to a previous report [20]. Erythema, scaling, and thickening were scored independently from 0 to 4 (0, none; 1, slight; 2, moderate; 3, marked; 4, very marked). The cumulative score (erythema score plus scaling score plus thickening score) served as the severity of inflammation index (scale 0–12). During PASI scoring, group allocation was known to the experimental coordinator to ensure proper treatment administration, while all clinical assessments were performed by an independent investigator blinded to group assignments to minimize bias.

### 2.10. Animal Study

A total of 45 C57BL/6 male mice were used across three independent experiments, with 5 mice per group in each experiment. In brief, C57BL/6 male mice were randomly assigned to three groups: Control group, IMQ + Vehicle group and IMQ + INCB3344 group [21,22]. Prior to the experiment, the hair on the backs of the mice was shaved and exposed to 2.5 cm × 3.5 cm rectangular hairless area. During a 5-day period, mice in the control group were smeared with 50 mg vaseline on the back skin every day, and mice in the other groups were smeared with 62.5 mg IMQ cream (Med·Shine, Chengdu, China; 41210301) on the dorsal skin every day [23]. Mice in the IMQ + Vehicle group or IMQ + INCB3344 group received intraperitoneal injections of vehicle or INCB3344 (30 mg/kg) (Selleck, Houston, TX, USA; S8220) every other day. Upon completion of the experiments, the mice were euthanized, and their skin was harvested for flow cytometry, RT-qPCR, and H&E staining.

### 2.11. Skin Cell Isolation and Flow Cytometry

Single cells from the dorsal skin were generated according to a previously reported study [24]. The antibodies for flow cytometry included APC-Cy7 Rat Anti-Mouse CD45(30-F11) (BD Pharmingen, Franklin Lakes, NJ, USA; 557659), BV711 Rat Anti-Mouse CD4(GK1.5) (BD Pharmingen, Franklin Lakes, NJ, USA; 563050), PerCP-Cy5.5 Rat Anti-Mouse CD8a(53-6.7) (BD Pharmingen, Franklin Lakes, NJ, USA; 551162), BV605 Hamster Anti-Mouse CD3e(145-2C11) (BD Pharmingen, Franklin Lakes, NJ, USA; 563004), Alexa Fluor 647 Rat anti-Mouse IL-17A (TC11-18H10) (BD Pharmingen, Franklin Lakes, NJ, USA; 560184); the Zombie Aqua™ Fixable Viability Kit (Biolegend, San Diego, CA, USA; 423102) was also used. The cells were analyzed using a FACS LSRFortessa or FACSymphony A3 flow cytometers (BD Biosciences, Franklin Lakes, NJ, USA). Data were analyzed using FlowJo v10 software and GraphPad Prism 9.

### 2.12. Statistical Analysis

All statistical analyses were performed using GraphPad Prism 9 software and are presented as the mean ± SEM. The results were analyzed using Student’s *t*-test (for two groups) or ANOVA (for more than two groups). A *p* value of <0.05 was considered to indicate statistical significance.

## 3. Results

### 3.1. Elevated Pyroptosis Levels in Psoriatic Lesions

To investigate the molecular characteristics and clinical relevance of pyroptosis in psoriasis, we conducted a cross-cohort analysis integrating 57 pyroptosis-related genes curated from the MsigDB database and literature mining (Appendix A) with 21 transcriptomic datasets of psoriasis from the GEO database spanning 2007–2020 (Appendix A). By employing a multi-dataset differential expression integration strategy (threshold: log2FC > 0.585, *p* ≤ 0.05), we identified 20 pyroptosis core genes that were consistently differentially expressed in at least five datasets (Figure 1A and Appendix AA), including key regulatory factors such as *AIM2*, members of the CASP family, and *IL1β*. Their significant upregulation suggests a crucial role of the pyroptosis pathway in psoriasis pathogenesis.

To elucidate the co-regulatory network of pyroptosis-related genes, we constructed a cross-dataset gene pair co-expression matrix. High-frequency co-occurrence analysis (correlation coefficient > 0.8; Figure 1B and Appendix AB) revealed a tightly interacting module comprising *AIM2*, *CASP5*, and *IL1β*. Through whole-genome correlation network integration and consensus clustering analysis (optimal cluster number k = 2, determined by the CDF curve; Appendix A), we categorized the 20 genes into two characteristic modules: Group 1 (*AIM2*, *BAK1*, *CASP1*, *CASP4*, *CASP5*, *GZMA*, *GZMB*, *IL1β*, *IRF1*, *NOD2*, *PYCARD*, *BAX*, *TNF*, *GSDMC*, *ZBP1*, and *NLRP7*) exhibited a highly correlated network, whereas Group 2 (*IL6*, *NLRP2*, *GSDME*, and *CYCS*) displayed a relatively independent expression pattern.

To validate the biological significance of these groupings, we performed GSVA using independent datasets with well-defined pyroptotic characteristics [25,26]. In the NLRC4-Macrophage Activation Syndrome (NLRC4-MAS)/Neonatal-Onset Multisystem Inflammatory Disease (NOMID) cohort (GSE57253), the enrichment score of Group 1 was significantly higher in the disease group than in controls (*p* = 0.04), closely aligning with the clinical pyroptotic state (Figure 1C). In the myocardial infarction time-series dataset (GSE153494), the Group 1 score peaked at 6 h post-injury (*p* = 0.028), accurately reflecting the dynamic changes in pyroptosis (Figure 1C). Conversely, Group 2 displayed heterogeneous variations across different disease models (Appendix A), suggesting potential tissue-specific regulatory mechanisms.

Further analysis demonstrated that the GSVA score of Group 1 could robustly distinguish psoriatic lesions from normal skin across all 21 psoriasis cohorts (AUC = 0.95; Figure 1D). Notably, following systemic treatment with ustekinumab or etanercept, the Group 1 score in psoriatic lesions significantly decreased (Figure 1E), confirming a positive correlation between pyroptosis levels and disease activity. In conclusion, through a multi-omics integrative approach, we first identify characteristic pyroptotic molecular modules in psoriasis, providing a theoretical foundation for targeting pyroptosis pathways as a therapeutic strategy.

### 3.2. MDMs Exhibit the Highest Pyroptosis Levels in Psoriatic Lesions

Next, we used CIBERSORT to estimate the infiltration abundance of various cell types in psoriatic lesion samples. Correlation analysis was performed to assess the association between pyroptosis status and immune cell infiltration. The results showed that classically activated macrophages (M1 macrophages) and activated dendritic cells exhibited a significant positive correlation with pyroptosis signature scores in at least five datasets, whereas resting mast cells and regulatory T cells were negatively correlated (Figure 2A). To further characterize cell type-specific pyroptotic features, we integrated a cross-tissue single-cell transcriptomic dataset developed by Gary Reynolds’ team, which includes both healthy skin and psoriatic lesions [17]. Using UMAP nonlinear dimensionality reduction, we identified 41 functional cell subpopulations within psoriatic lesions, comprising 17 stromal cell types, 14 antigen-presenting cell types, and 10 lymphocyte/mast cell types (Figure 2B). Pyroptotic activity was quantified across these subpopulations using the AUCell algorithm, which demonstrated comparable effectiveness to GSVA scores in reflecting pyroptosis states (subtype-level Spearman’s ρ = 0.774, *p* < 0.001; Appendix A). After threshold optimization (optimal threshold = 0.0864, determined by AUCell explore thresholds, with a Youden index of 0.72), we found that the majority of cell subtypes exhibited low pyroptotic activity. However, MDMs, inflammatory macrophages, and monocyte-derived dendritic cells contained a relatively higher proportion of pyroptotic cells, with MDMs exhibiting the highest proportion of pyroptotic cells (Figure 2C). To independently validate this finding, we analyzed a second scRNA-seq dataset (GSE228421). After rigorous quality control and reclustering of myeloid subsets (Appendix A), MDMs again exhibited significantly higher pyroptosis AUCell scores than other cell types (Appendix A). This dual-dataset confirmation solidifies MDMs as the dominant pyroptosis-executing cells in psoriatic lesions.

To experimentally validate whether MDMs in psoriasis patients exhibit a high pyroptotic state, PBMCs were isolated from psoriasis patients and healthy controls and subsequently induced into macrophages in vitro (Figure 2D). Western blot analysis showed that psoriasis-derived MDMs had elevated expression of pyroptosis-related proteins, Caspase 1 and Caspase 4, when compared to healthy controls (Figure 2E,F and Appendix A). Furthermore, RT-qPCR analysis of pyroptosis-associated inflammatory cytokines revealed significantly increased expression of *IL1β*, *IL6*, and *TNFα* in psoriasis-derived MDMs compared to healthy controls (Figure 2G and Appendix A). These findings collectively suggest that MDMs derived from psoriasis patients exhibit a high pyroptotic state.

### 3.3. Highly Pyroptotic MDMs Drive Psoriasis Progression Through CXCL16

To investigate the impact of the pyroptotic state of MDMs on the immune microenvironment in psoriasis, we performed functional profiling of MDMs based on AUCell scores (high/low pyroptosis groups) and constructed an intercellular communication network using the CellChat algorithm. Comparative analysis revealed that both high and low pyroptotic MDMs were enriched in signaling output pattern 1 (Table 1). However, the high pyroptosis group specifically activated the CXC chemokine (CXCL) pathway (Table 1), suggesting that these cells may mediate inflammatory cascades through CXCL signaling. Further ligand-receptor contribution analysis (net analysis contribution) revealed that the CXCL pathway consists of seven pairs of core molecules, including five ligands (CXCL1, CXCL3, CXCL5, CXCL12, and CXCL16) and four receptors (ACKR1, ACKR3, CXCR4, and CXCR6) (Appendix A). Single-cell resolution expression profiling showed that high pyroptotic MDMs predominantly expressed high levels of CXCL1, CXCL3, CXCL5, and CXCL16 (Figure 3A and Appendix AB). Visualization of the ligand-receptor interaction network revealed that high pyroptotic MDMs transmit chemotactic signals to innate lymphoid cells 2 (ILC2), Tc17, Th17, and Treg cells via the CXCL16-CXCR6 axis, while also activating endothelial cells through the CXCL1/3/5-ACKR1 pathway (Figure 3B,C), thereby forming an intercellular inflammatory regulatory network.

To validate the clinical relevance of the CXCL pathway, we performed a combined expression profile analysis of chemokines in psoriatic lesional tissues across 21 GEO datasets. The results showed that *CXCL1* and *CXCL16* were significantly upregulated across all datasets in psoriatic lesions (Figure 3D,E), while *CXCL3* and *CXCL5* were only correlated in a subset of datasets (Appendix A). RT-qPCR experiments further confirmed that *CXCL16* expression was significantly elevated in MDMs from psoriasis patients compared to healthy controls, while no statistical differences were observed for *CXCL1*, *CXCL3*, and *CXCL5* (Figure 3F and Appendix A). Subsequently, to determine whether CXCL16 expression levels were causally regulated by pyroptosis induction, we treated psoriasis-derived MDMs with the pyroptosis-specific inhibitor VX-765 for 48 h and quantified CXCL16 gene expression. This intervention revealed a significant downregulation of *CXCL16* in VX-765-treated cells compared to untreated controls (Figure 4G and Appendix A), thus demonstrating that CXCL16 expression is directly modulated by pyroptotic activity in MDMs. Collectively, these findings suggest that CXCL16 may be a central molecule driving the inflammatory output of high pyroptotic MDMs in psoriasis.

### 3.4. Inhibition of MDMs Can Alleviate IMQ-Induced Psoriatic Dermatitis

To further investigate whether inhibiting MDM recruitment could ameliorate psoriasis pathology, we employed a CCR2 antagonist in an IMQ-induced murine psoriasiform dermatitis model. This approach was based on the critical role of CCL2-CCR2 axis in recruiting CCR2^+^ monocytes to inflammatory lesions where they differentiate into macrophages. CCR2 inhibitor INCB3344 (30 mg/kg, administered every other day) was administered orally starting from day 0, concurrent with daily topical IMQ application for 4 days. Mice were sacrificed on day 5 for tissue collection. The INCB3344 + IMQ group showed marked clinical improvement compared to IMQ + Vehicle controls, characterized by reduced erythema, decreased scaling, and significantly lower PASI scores (Figure 4A,B), without observable toxicity, as evidenced by comparable body weights between groups (Figure 4C). Histopathological analysis revealed significant epidermal thinning in INCB3344-treated mice (Figure 4D,E). Flow cytometry demonstrated significant reductions in both M1 macrophage and Th17 cell populations within lesions of the treatment group (Figure 4F,I and Appendix AA,B), while Tc17 cell proportions remained unchanged (Figure 4J). RT-qPCR analysis confirmed downregulation of psoriasi-associated genes (*Il17a*, *Il17c*, *Il17f*, *Il22*, *Il1β*, *Tnfα*, *Il6*, and *Cxcl16*) in INCB3344-treated mice, consistent with attenuated inflammatory responses (Figure 4K,L).

## 4. Discussion

Psoriasis is a chronic inflammatory skin disease with pathogenesis closely related to abnormal activation of the immune system, particularly the IL-17/IL-23 axis, activation of Th17 cells, macrophages, and dendritic cells [1]. Pyroptosis, a pro-inflammatory programmed cell death pathway, exacerbates inflammatory responses through the release of cytokines such as IL-1β and IL-18 [6]. In this study, we screened 57 pyroptosis-related genes from the MsigDB database and literature and performed cross-cohort analysis using 21 psoriasis lesional transcriptomic datasets from the GEO database. We found that the pyroptotic state was elevated in psoriasis patients compared to healthy controls and was associated with treatment response, suggesting that pyroptosis plays an important role in psoriasis pathogenesis and is proposed as a viable target for pharmaceutical intervention. In terms of mechanism, previous studies have shown that in psoriasis, NLRP3 may recognize microbial components (e.g., lipopolysaccharides) or endogenous danger signals (e.g., urate crystals, reactive oxygen species), activate caspase-1, and cleave GSDMD, leading to pyroptosis and IL-1β release, thereby amplifying skin inflammation [6,27]. GSDMD/E, as an executioner protein of pyroptosis, may be aberrantly activated in psoriasis lesions, promoting keratinocyte pyroptosis and exacerbating epidermal hyperplasia and inflammatory infiltration [8,10,13]. In terms of clinical transformation, Lai et al. found that Gsdmd knockdown suppressed pyroptosis and improved skin lesion severity in the IMQ-induced psoriasis-like mouse model [10]. Li et al. reported that a caspase-3 inhibitor could reduce IMQ-induced psoriasis-like skin inflammation in mice [8]. Canakinumab, targeting IL-1β, has been used in the treatment of pustular psoriasis [28]. However, the majority of studies on pyroptosis as a potential therapeutic target are still in the primary research stage.

However, the main cell types and specific functional mechanisms of pyroptosis in psoriasis remain unclear. In this study, we constructed a single-cell transcriptomic atlas of psoriasis and conducted pyroptosis-related cell infiltration analysis, identifying MDMs as the cell type with the highest pyroptotic levels in psoriasis. A previous study also revealed that macrophages from psoriatic skin exhibited higher pyroptosis gene set scores compared to control skin cells, while keratinocytes showed lower pyroptosis gene set scores [29]. Nevertheless, we acknowledge that other cell types likely contribute to pyroptosis-driven inflammation. Keratinocytes, for instance, have been implicated in psoriasis pathogenesis through GSDMD/E-mediated pyroptosis [8,13]. Future studies should dissect cell-type-specific pyroptotic mechanisms across diverse immune and stromal compartments to fully elucidate their cooperative roles in disease progression.

Macrophages are typically classified into MDMs and tissue-resident macrophages based on their origin, and into pro-inflammatory M1 macrophages and anti-inflammatory M2 macrophages based on their function. In psoriasis, studies suggest that macrophages can participate in disease progression through pyroptosis-related NLRP3 inflammasome activation [30,31]. Besides psoriasis, macrophage pyroptosis is involved in various diseases, such as pregnancy-induced gut microbiota changes that drive macrophage pyroptosis and exacerbate septic inflammation [32], hyperglycemia-associated macrophage pyroptosis accelerating periodontal inflamm-aging [33], and macrophage pyroptosis interacting with mitophagy to regulate mitochondrial homeostasis in pulmonary fibrosis [34]. However, these studies do not specifically address whether MDM-mediated pyroptosis is involved in disease progression.

Previous studies have indicated that pyroptosis in psoriasis triggers caspase-1 activation, increases IL-1β and IL-18 expression, and activates the IL-23/Th17 pathway, leading to the secretion of large amounts of inflammatory cytokines and chemokines, which ultimately induces skin inflammation [35]. In this study, we integrated multi-omics analysis and single-cell resolution functional analysis, revealing that highly pyroptotic MDMs may mediate signaling through the CXCL pathway to Th cells, Tc cells, Treg cells, ILCs, and endothelial cells. Cross-cohort transcriptomic analysis and in vitro experiments further showed that CXCL16 (rather than CXCL1/3/5) is the core effector molecule in highly pyroptotic MDMs. Nevertheless, whether non-pyroptotic macrophage subsets employ distinct chemokine networks (e.g., CCL20-CCR6) warrants further investigation. CXCL16 is a CXC soluble chemokine, adhesion molecule, and cell surface clearance receptor that regulates inflammation, tissue damage, and fibrosis [36]. Previous studies have shown that in psoriasis, the CXCR6 axis induces CD8^+^ T cell migration to human skin and promotes the recruitment of T lymphocytes in joint cavities [37]. Hornburg et al. found that infiltrating tumor cells highly express CXCL16, attracting CXCR6^+^ T cells to the tumor epithelial region [38]. Piehl et al. found that in cerebrospinal fluid from patients with cognitive impairment, CXCL16 secretion increased, recruiting CD8^+^ T cells via CXCR6, which, upon activation, released granzyme and other effector molecules, exacerbating neuroinflammation and axonal damage [39]. Therefore, the CXCL16/CXCR6 axis may be a potential therapeutic target for tumors [38,40,41,42], kidney diseases [36,42], cardiovascular diseases [36,42], non-alcoholic fatty liver disease 43, and autoimmune diseases [37]. While our data implicate highly pyroptotic MDMs in Th17 activation via CXCL16-CXCR6, formal causal validation requires future lineage-specific ablation of pyroptosis executors (e.g., Gsdmd).

This study integrates multi-omics approaches to reveal the pyroptotic characteristics of psoriasis lesions and suggests that highly pyroptotic MDMs may participate in psoriasis pathogenesis through the CXCL16/CXCR6 signaling axis. However, there are several limitations: First, the single-cell data analysis is based on a single cohort with a small sample size and does not encompass differences across disease stages (such as active and remission phases), and most public datasets did not report detailed demographic/ethnic diversity, which may limit the generalizability of the conclusions. Future multi-center studies with larger, diverse cohorts and dynamic sampling across disease phases are needed to validate pyroptosis dynamics in MDMs and CXCL16-mediated mechanisms. Second, although the AUCell threshold method defines pyroptosis status with statistical significance, it lacks direct validation through in vivo and in vitro functional experiments (e.g., CRISPR knockout of pyroptosis-related genes) to establish a causal relationship between MDMs and CXCL pathway activation. Third, the downstream mechanisms of the CXCL16-CXCR6 axis remain unclear, such as whether it affects Th17 activation via the JAK-STAT or NF-κB pathways, which requires further exploration. Lastly, although RT-qPCR confirmed increased CXCL16 expression in MDMs from psoriasis patients, there is a lack of tissue-based spatial transcriptomics or protein-level validation, and the efficacy of CXCL16 blockers in preclinical models has not been evaluated. Future studies should expand cohort sizes, establish conditional gene-editing models, and develop targeted intervention strategies to facilitate clinical translation.

## 5. Conclusions

In this study, by integrating multi-omics analysis and single-cell resolution functional analysis, we demonstrated that the level of pyroptosis in psoriasis patients is elevated and positively correlated with treatment efficacy, suggesting its utility as a predictive biomarker in clinical settings. Furthermore, we revealed for the first time the pivotal role of pyroptosis in MDMs in reshaping the immune microenvironment in psoriasis. Cross-cohort transcriptome analysis combined with in vitro validation experiments showed that CXCL16 is significantly upregulated in psoriatic MDMs, suggesting that CXCL16 may serve as a critical molecular switch in the inflammatory cascade mediated by hyperpyroptotic MDMs. Hyperpyroptotic MDMs may promote the chemotaxis and activation of Th17 cells via specific activation of the CXCL16-CXCR6 signaling axis, thereby facilitating the onset and progression of psoriasis. This study not only elucidates the functional coupling mechanism between the pyroptosis characteristic of MDMs and the CXCL pathway but also provides a novel theoretical basis and potential therapeutic target for psoriasis treatment by targeting the pyroptosis-CXCL16 axis.

## Figures and Tables

**Figure 1 biomedicines-13-01763-f001:**
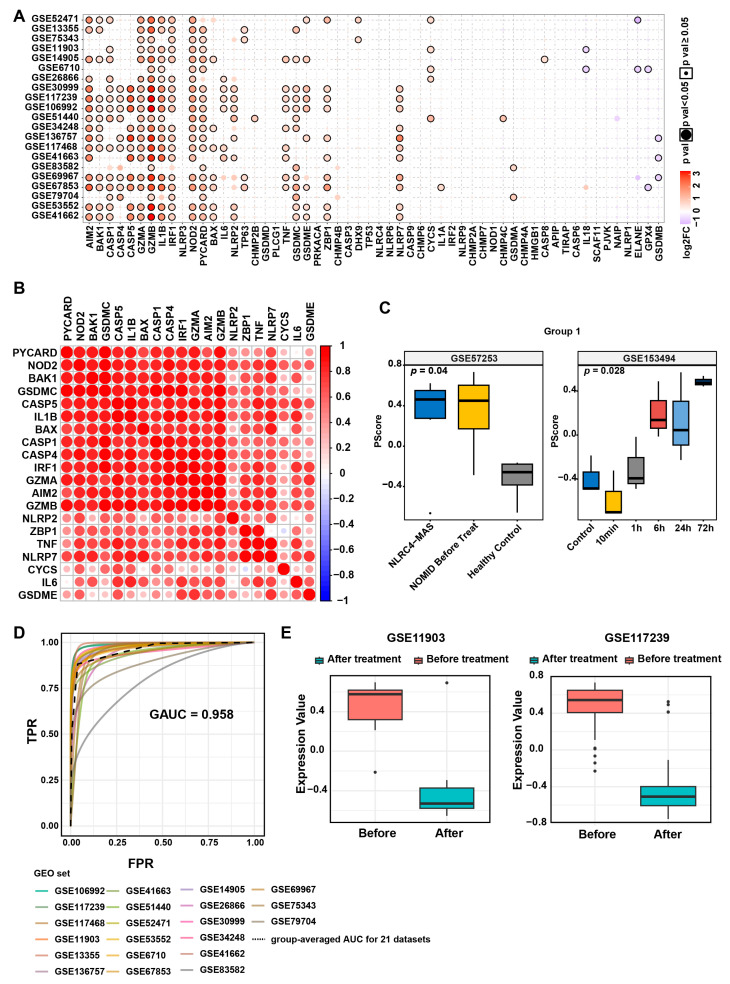
Association of pyroptosis with psoriasis and development/validation of the pyroptosis score. (**A**) Dot plot showing the expression differences of pyroptosis-related genes between psoriasis and normal controls across bulk-RNA datasets. (**B**) Dot plot illustrating the expression patterns of 20 pyroptosis-related genes in the dataset, as determined by the non-negative matrix factorization (NMF) algorithm. (**C**) Boxplot validating the pyroptosis score in the pyroptosis-related datasets GSE57253 and GSE153494. (**D**) Average ROC curve evaluating the discriminatory power of the pyroptosis score in psoriasis. (**E**) Boxplot demonstrating changes in the pyroptosis score before and after treatment.

**Figure 2 biomedicines-13-01763-f002:**
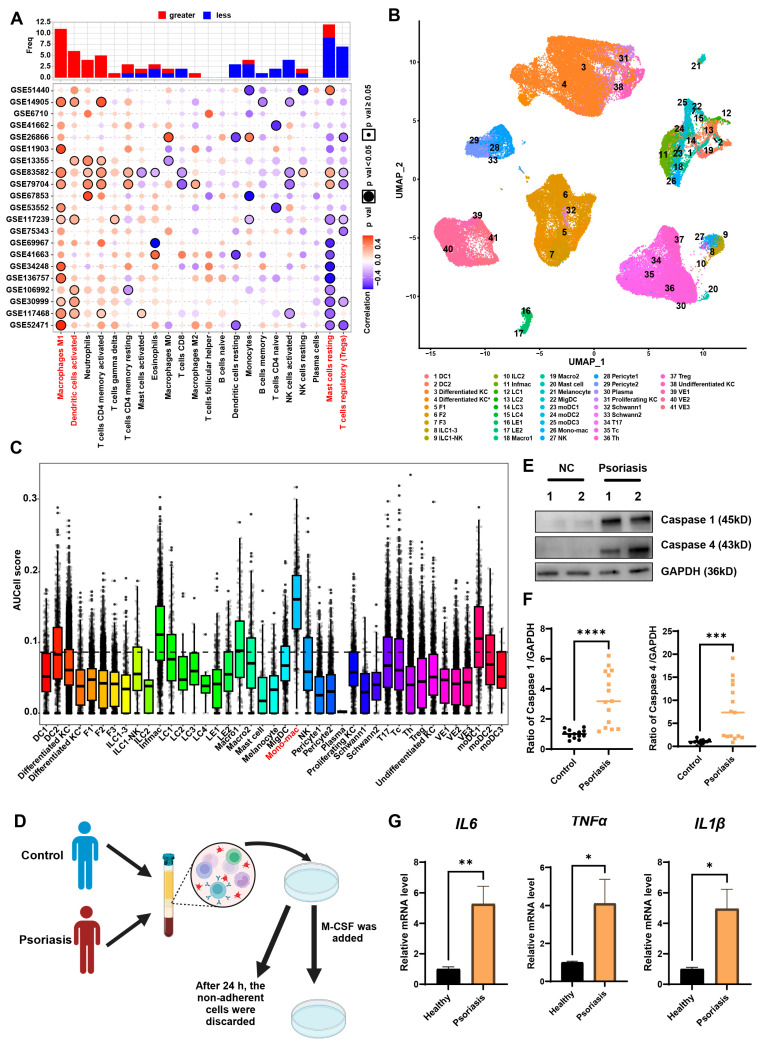
MDMs exhibit elevated pyroptosis levels in psoriasis. (**A**) Dot plot displaying the association between immune cells and pyroptosis scores, as analyzed by CIBERSORT immune enrichment. (**B**) UMAP visualization of cell subtype distribution in psoriasis single-cell data. (**C**) Boxplot comparing pyroptosis scores across different cell subtypes. (**D**) Flowchart of extracting PBMCs from human peripheral blood and inducing them into MDMs. (**E**,**F**) Representative immunoblotting images and analysis of Caspase 1 and Caspase 4 expression in MDMs from healthy controls and psoriasis patients (Control = 13, Psoriasis = 15). (**G**) Quantitative PCR analysis of *Il1β*, *Tnfα* and *Il6* in MDMs from healthy controls and psoriasis patients (*n* = 5–10/group). Error bars show the mean ± SEM. *, *p* < 0.05; **, *p* < 0.01; ***, *p* < 0.001; ****, *p* < 0.0001. The *p* value was determined using two-tailed unpaired Student’s *t*-test (**F**,**G**). Data are representative of three independent experiments. Mono-mac, monocyte-derived macrophages.

**Figure 3 biomedicines-13-01763-f003:**
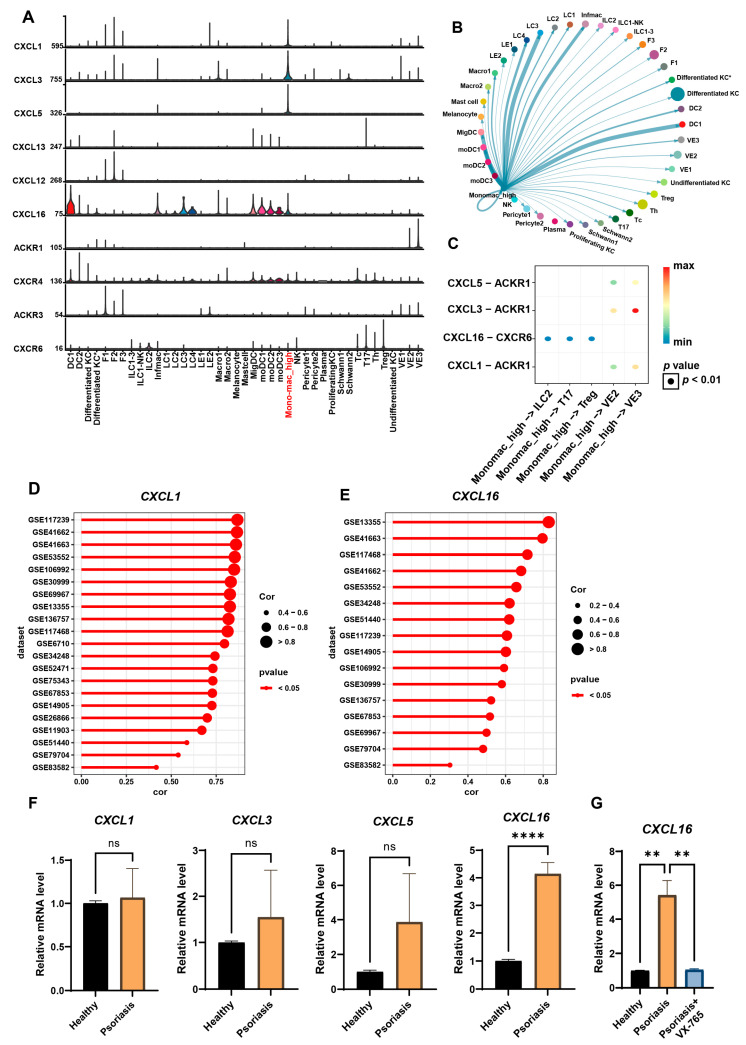
High-pyroptosis MDMs mediate intercellular signaling primarily through the CXCL1/3/5/16 axis. (**A**) Violin plots showing CXCL pathway activity across cell subtypes, as revealed by cell-cell communication analysis. (**B**) Cell communication map between hyperpyroptotic MDMs and other types of cells in the skin. (**C**) Dot plot illustrating outgoing signals from high-pyroptosis MDMs via CXCL pathways. (**D**) Correlation between CXCL1 expression levels and psoriasis severity across bulk-RNA datasets. (**E**) Correlation between CXCL16 expression levels and psoriasis severity across bulk-RNA datasets. (**F**) Quantitative PCR analysis of *CXCL1/3/5/16* in MDMs from healthy controls and psoriasis patients (*n* = 3–10/group). (**G**) Quantitative PCR analysis demonstrating significant downregulation of CXCL16 mRNA in VX-765-treated MDMs (*n* = 3/group). Error bars show the mean ± SEM. **, *p* < 0.01; ****, *p* < 0.0001; ns, not significant. The *p* value was determined using a two-tailed unpaired Student’s *t*-test for (**F**,**G**). Data are representative of three independent experiments. Mono-mac, monocyte-derived macrophages.

**Figure 4 biomedicines-13-01763-f004:**
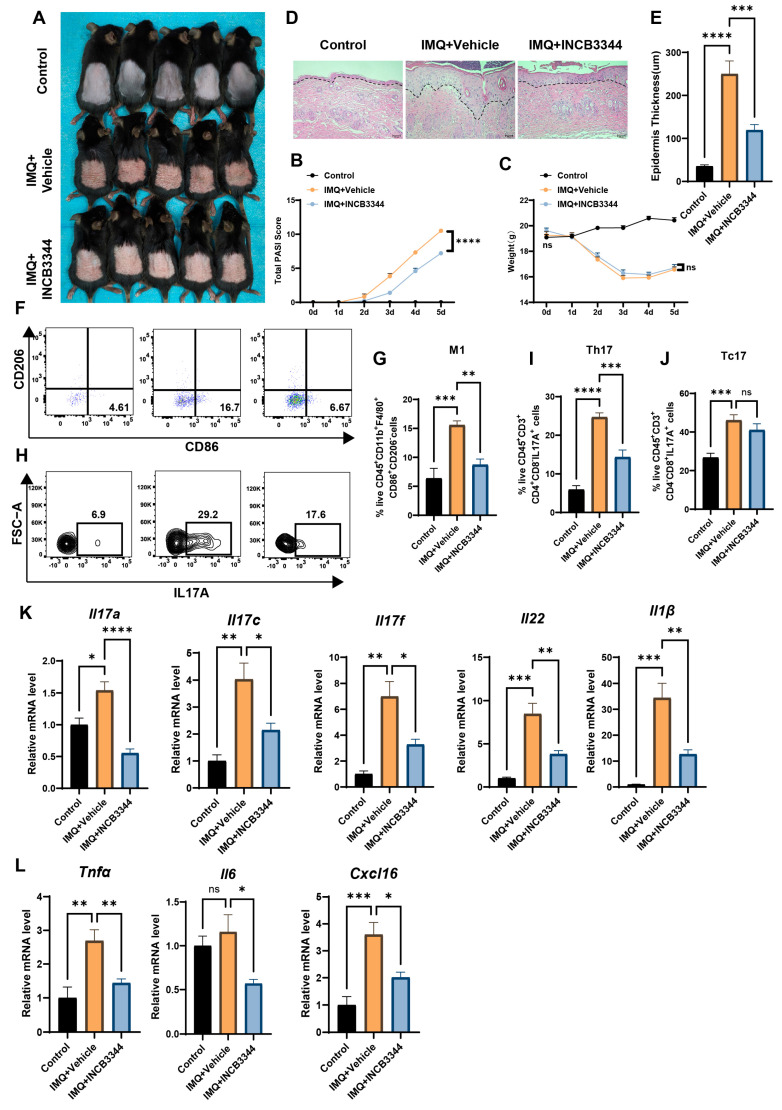
Inhibition of MDMs improves IMQ-induced psoriasis-like dermatitis. (**A**) Images of the dorsal back from Control, IMQ + Vehicle and IMQ + INCB3344 mice (*n* = 5/group) on day 5. (**B**) Daily scoring of the back skin in Control, IMQ + Vehicle and IMQ + INCB3344 mice (*n* = 5/group). (**C**) Daily weight in Control, IMQ + Vehicle and IMQ + INCB3344 mice (*n* = 5/group). (**D**,**E**) Representative images of back skin stained by hematoxylin and eosin (**D**) and statistical analysis of epidermal thickness measurements (**E**) (*n* = 5/group). Scale bars, 100 μm. (**F**,**G**) Flow cytometry analysis of M1 macrophages in the back skin of mice (*n* = 5/group). (**H**,**I**) Flow cytometry analysis of Th17 cells in the back skin of mice (*n* = 5/group). (**J**) Flow cytometry analysis of Tc17 cells in the back skin of mice (*n* = 5/group). (**K**,**L**) Quantitative PCR analysis of *Il17a*, *Il17c*, *Il17f*, *Il22*, *Il1β*, *Tnfα*, *Il6*, *Cxcl16*) from the back skin of Control, IMQ + Vehicle and IMQ + INCB3344 mice (*n* = 3–5/group). Error bars show the mean ± SEM. *, *p* < 0.05; **, *p* < 0.01; ***, *p* < 0.001; ****, *p* < 0.0001; ns, not significant. The *p* value was determined using one-way ANOVA (**B**,**C**,**E**,**G**,**I**–**L**). Data are representative of three independent experiments.

**Table 1 biomedicines-13-01763-t001:** Comparison of MDMs efferent signaling in high- and low-pyroptosis states.

Signaling Pathways	Signaling Pattern	High-Pyroptosis MDMs	Low-Pyroptosis MDMs
*CXCL*	signaling pattern 1	√	×
*VISFATIN*	signaling pattern 1	√	√
*MHC-II*	signaling pattern 1	√	√
*IGTB2*	signaling pattern 1	√	√
*IL1*	signaling pattern 1	√	√
*EGF*	signaling pattern 1	√	√
*NECTIN*	signaling pattern 1	√	√
*TNF*	signaling pattern 1	√	√
*CD86*	signaling pattern 1	√	√
*CD39*	signaling pattern 1	√	√
*CD80*	signaling pattern 1	√	√
*IL10*	signaling pattern 1	√	√
*ALCAM*	signaling pattern 1	√	√
*SEMA7*	signaling pattern 1	√	√
*SEMA4*	signaling pattern 1	√	√

√: Pathway present in the cell subtype; ×: Pathway absent.

## Data Availability

The dataset used in this study is publicly available in a publicly accessible repository (https://www.ncbi.nlm.nih.gov/geo/; https://www.gsea-msigdb.org/gsea/msigdb; https://zenodo.org/records/4569496).

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
