# Peer review of "Integrated Multi-Omics Profiling Reveals That Highly Pyroptotic MDMs Contribute to Psoriasis Progression Through CXCL16"

_biomedicines, 2025, doi:10.3390/biomedicines13071763_

Round 1
Reviewer 1 Report
Comments and Suggestions for Authors
The original article "Single-Cell Multi-Omics Integration Analysis Reveals that High Pyroptotic MDMs Contribute to Psoriasis Progression through CXCL16" by Liping Jin et al. revealed that MDMs in psoriatic lesions exhibit a hyperactive pyroptotic state, which contributes to disease progression through CXCL16-mediated remodeling of the immune microenvironment. This is a novel and relevant contribution to the psoriatic research community.
Regarding the research design, it is appropriate, and the methods are adequately described. Regarding the manuscript and the supporting information, they are well-written and clear, and the English level is adequate and does not require any improvement. The introduction provides sufficient background about psoriasis, pyroptosis, and MDMs; the results, figures, and tables are clearly presented, and the conclusions are supported by the results.
According to the above-mentioned aspects, my overall recommendation is that the article should be accepted for publication in Biomedicines in its present form. I do not have further suggestions for the authors.
Author Response
We are delighted to receive the positive feedback and recommendation for acceptance from the reviewer. We sincerely appreciate the reviewer’s recognition of the novelty and relevance of our findings regarding hyperactive pyroptotic MDMs and their role in psoriasis progression through CXCL16. We are also grateful for the reviewer’s comments affirming the appropriateness of the research design, clarity of the manuscript, and the sufficiency of the background and results presentation. We thank the reviewer for their time and valuable assessment of our work.
Reviewer 2 Report
Comments and Suggestions for Authors
The study reveals that pyroptosis in monocyte-derived macrophages (MDMs) significantly contributes to psoriasis pathogenesis, particularly through CXCL16-mediated Th17 activation. Researchers analyzed 21 bulk transcriptomic datasets and single-cell RNA-seq data to identify two pyroptosis-related gene modules, with Group 1 strongly associated with disease severity and therapeutic response. The study validated key findings in vitro and in vivo, strengthening clinical relevance. However, limitations include the complexity of data presentation and a narrow focus on MDMs without exploring other immune or stromal cell types contributing to pyroptosis-driven inflammation. The study also highlights CXCL16/CXCR6 signaling as a mechanistic pathway, but deeper mechanistic dissection could further support causal claims.
The study lacks comprehensive functional validation, focusing mainly on monocyte-derived macrophages, which could be expanded to include other immune or non-immune cell types. Cross-platform integration challenges arise from the integration of multiple datasets over a decade, potentially introducing batch effects or inconsistencies in data preprocessing. Population and cohort diversity is also a concern, as most datasets lack demographic or ethnic diversity reporting. Expanded validation of CXCL16 is needed, as it is only one of several chemokines likely involved in macrophage-T-cell interactions. The single-cell data lacks longitudinal or post-treatment samples, which would better capture pyroptosis dynamics in response to therapy. The clinical translation gap is still conceptual, and future studies should explore whether pyroptosis markers predict treatment response or correlate with disease severity in clinical settings. Despite these limitations, the study identifies pyroptosis as a dynamic and targetable inflammatory driver in psoriasis, offering a promising avenue for precision immunotherapy.
Reviewer 3 Report
Comments and Suggestions for Authors
- Although the introduction comprehensively reviews the background of pyroptosis and psoriasis, the study lacks a clearly formulated hypothesis that explicitly defines the proposed relationship between pyroptotic MDMs and psoriasis progression through the CXCL16 axis. The authors should clearly state a central hypothesis that connects pyroptosis, MDM activity, and CXCL16-mediated immune modulation.
- The single-cell analysis relies on a single cohort, which raises concerns regarding the reproducibility and generalizability of the findings across diverse patient populations and disease stages. The study should incorporate multiple independent single-cell datasets to validate the pyroptotic signatures and cell type distributions observed.
- While the AUCell scoring method is statistically appropriate for defining pyroptosis status, the study lacks functional experiments to establish a causal relationship between pyroptotic MDMs and CXCL16 pathway activation.
Genetic manipulation (e.g., CRISPR-Cas9 knockout of pyroptosis-related genes) or pharmacological inhibition of pyroptosis should be performed to validate the proposed mechanism. - The study does not elucidate the downstream signaling pathways through which the CXCL16-CXCR6 axis may influence Th17 differentiation and immune cell recruitment. Future work should investigate whether the JAK-STAT or NF-κB pathways are involved in CXCL16-mediated Th17 activation to provide deeper mechanistic insights.
- Although RT-qPCR confirmed elevated CXCL16 expression in psoriasis-derived MDMs, protein-level confirmation and tissue-based localization (e.g., immunohistochemistry, ELISA, or spatial transcriptomics) are missing.
The authors should provide protein expression data and spatial validation to confirm the functional relevance of CXCL16 in the lesional tissue microenvironment. - The study identifies CXCL16 as a potential effector but does not evaluate the therapeutic impact of CXCL16 inhibition in preclinical models.
The authors should consider testing CXCL16-targeting strategies (e.g., neutralizing antibodies or small molecule inhibitors) to assess the therapeutic potential of disrupting this pathway. - The sample sizes for both patient-derived cells and animal experiments are relatively small (n=3–15), potentially affecting the statistical robustness of the findings.Expanding the sample size and ensuring balanced demographic representation would strengthen the study's validity and reproducibility.
- The manuscript does not report a priori power calculations to justify the selected sample sizes. The authors should perform and report a statistical power analysis to confirm that the study is adequately powered to detect meaningful differences.
- There are minor terminological inaccuracies and typographical errors throughout the manuscript (e.g., "hyperpyretic MDMs" should be "hyperpyrotic MDMs"). A thorough language and technical revision is needed to ensure consistency and precision in scientific terminology.
Overall, this study presents a novel and potentially impactful finding regarding the role of pyroptotic MDMs and the CXCL16-CXCR6 axis in psoriasis pathogenesis. However, the study currently requires major revisions to address the limitations in experimental validation, dataset diversity, and mechanistic exploration. Addressing these issues will substantially strengthen the manuscript's scientific rigor and translational relevance.
Reviewer 4 Report
Comments and Suggestions for Authors
1- The abstract states that 21 transcriptomic datasets were integrated, but fails to mention how batch effects and platform heterogeneity were addressed.
2- The use of both GSVA and AUCell to quantify pyroptosis activity needs justification.
3- While single-cell RNA-seq is cited, the level of cell-type resolution (e.g., clustering method, marker validation) is not mentioned.
4-The ms implies that pyroptosis “drives” inflammation, yet causal links between pyroptosis activation and Th17 induction are not adequately supported without loss-of-function studies.
5- The CXCL16-CXCR6 signaling claim lacks direct evidence, no mention of receptor/ligand blockade, knockdown, or signaling assays to confirm the interaction’s functional role.
6-The imiquimod-induced model is mentioned, but treatment duration, dosing, and scoring methods for inflammation are not specified.
7- While pyroptosis is proposed as a therapeutic target, no specific inhibitors or translational strategies are discussed to bridge the gap between mechanistic insight and clinical application.
8-The complex plasticity of macrophage subtypes is reduced to M1/M2 dichotomy, which may be overly simplistic for psoriasis pathogenesis.
9- The dynamics of pyroptosis induction during disease progression or therapy response are not explored.
Reviewer 5 Report
Comments and Suggestions for Authors
The manuscript is devoted to the current problem - the assessment of the mechanisms of pyroptosis and the role of macrophages in the pathogenesis of the inflammatory response in psoriasis. The results of studying the mechanisms of the inflammatory response obtained in the work give the authors grounds to outline promising ways of creating drugs based on CCR2 inhibitor. At the same time, there are some shortcomings that need correction or additional explanations.
1. The authors focused their attention on pyroptosis and the role of macrophages as inducers of the inflammatory response - the promoter of pyroptosis. However, it is known that the inflammatory component of psoriasis plaques is primarily due to neutrophil infiltration into the epidermis, forming neutrophil abscesses such as the spongiform pustule of Kogoj and Munro's microabscess.
2. In continuation of the first remark, the question arises about the adequacy of the mouse IMQ model of psoriasis. Although this model is used, it is not clear to what extent it reflects the autoimmune nature of the disease and, accordingly, the nature of leukocyte infiltration.
3. The authors study the mechanism of cell death in plaques, while in psoriasis, on the contrary, there is a pronounced proliferation of keratinocytes.
4. The studies of peripheral blood MDMs from healthy controls and psoriasis patients are not entirely clear. What can the differences mean? Are they a manifestation of a systemic reaction of local inflammation? How do these data reflect the phenomenon of pyroptosis in plaques?
5. The text of the manuscript is difficult to read, since it contains various sections that are weakly related to each other. Additionally, the construction and logic need to be adjusted.
6. The results of the study do not fully correspond to the title, since not only Single-Cell Multi-Omics Analysis was performed.
Round 2
Reviewer 4 Report
Comments and Suggestions for Authors
Accepted for publication
Reviewer 5 Report
Comments and Suggestions for Authors
The authors have addressed my concerns and changed the title.
Comments on the Quality of English Language
Once all the edits have been accepted, it should be read again for grammatical errors.